# Lime and Gypsum Rates Effects in New Soybean Areas in the Cerrado of Matopiba, Brazil

Doze Batista de Oliveira [1,*], Julian Junio de Jesus Lacerda [1], Adenilson Pereira Cavalcante [1], Karmem Guimarães Bezerra [1], Allana Pereira Moura da Silva [1], Ana Caroline Guimarães Miranda [1], Tiago Pieta Rambo [1], Rafael Maschio [2], Hosana Aguiar Freitas de Andrade [3], Paula Muniz Costa [3], Carlos Antonio Ferreira de Sousa [4], José Oscar Lustosa de Oliveira Júnior [4], Edvaldo Sagrilo [4] and Henrique Antunes de Souza [4]

1   Federal University of Piauí (UFPI), Campus Professora Cinobelina Elvas (CPCE), Bom Jesus 64900-000, PI, Brazil; julian@ufpi.edu.br (J.J.d.J.L.); adepereira@ufpi.edu.br (A.P.C.); karmem.feedera@ufpi.edu.br (K.G.B.); allana_moura@ufpi.edu.br (A.P.M.d.S.); carolmiranda@ufpi.edu.br (A.C.G.M.); tiagopieta@hotmail.com (T.P.R.)
2   Association of Soybean and Corn Producers of Piauí (APROSOJA-PI), Bom Jesus 64049-250, PI, Brazil; diretoriaexecutiva@aprosojapi.com.br
3   Agrarian Sciences Center (CCA), Federal University of Piauí (UFPI), Teresina 64049-550, PI, Brazil; hosana.andrade@ufpi.edu.br (H.A.F.d.A.); paula.costa@ufpi.edu.br (P.M.C.)
4   Brazilian Agricultural Research Company, Embrapa Meio-Norte, Teresina 64008-780, PI, Brazil; carlos.antonio@embrapa.br (C.A.F.d.S.); jose.oscar@embrapa.br (J.O.L.d.O.J.); edvaldo.sagrilo@embrapa.br (E.S.); henrique.souza@embrapa.br (H.A.d.S.)
*   Correspondence: doze@ufpi.edu.br

**Abstract:** High rates of limestone have been increasingly utilized in newly converted areas for grain production in agricultural frontier regions to expedite the short-term correction of soil fertility, leading to compensatory yields. However, there is a lack of information about different doses of lime and gypsum for soils in the Cerrado of Matopiba, especially in the state of Piauí, Brazil. The aim of this study was to evaluate the effects of doses of lime and gypsum in newly converted areas for soybean production in the Cerrado of Southwest Piauí. The study was carried out in the 2019/2020 and 2020/2021 crop years, on yellow Oxisol soil, in a randomized block design and treatments following a $5 \times 4$ factorial: five lime rates (0, 5, 10, 15, and 20 t ha$^{-1}$) and four gypsum rates (0, 1, 2 and 4 t ha$^{-1}$), with four replicates. The standard lime and gypsum rates were 5 t ha$^{-1}$ and 1 t ha$^{-1}$, respectively. Soil fertility attributes (0.0–0.2, 0.2–0.4, and 0.4–0.6 m), nutritional status of plants, and soybean yield were measured. The increases in grain yield using a lime rate of 10 t ha$^{-1}$ were 18% and 12% in the 2019/2020 and 2020/2021 crop years, respectively. High lime rates provide a reduction in the concentrations of P, K, and cationic micronutrients in soil, thereby reducing leaf contents of macro- and micronutrients in soybean plants. Concentrations of Ca, Mg, and S in subsurface layers were raised to proper levels, similar to those recommended for topsoil (0.0–0.2 m). The use of gypsum and lime in newly converted areas for soybean cultivation provides quick improvement in soil chemical conditions and reduction in acidity components. The application of 10 t ha$^{-1}$ of lime improved the soil chemical environment in the Matopiba region the short time available for chemical reactions to occur, allowing soybean cultivation in newly converted areas of Cerrado into agriculture.

**Keywords:** *Glycine max*; soil correction; soil fertility; aluminum neutralization; correction speed

## 1. Introduction

Soybeans play an important role in Brazilian agriculture, accounting for 49.15% of grain production in the country [1]. In the 2022/2023 crop year, soybean production in Brazil was approximately 153.6 million tons from a cultivated area of 43.5 million hectares [1]. The country's production represents around 41% of global soybean production [2]. Soybean cultivation has been expanding most intensely into the Cerrado of a

region known as Matopiba. Matopiba consists of 337 municipalities, distributed over a total area of 73 million ha [3] in the Brazilian states of Maranhão, Tocantins, Piauí, and Bahia. Soybeans are grown on an area of approximately 5.13 million hectares, which is equivalent to 12.5% of Brazil's soybean production area [4].

The region strategic region for the country's economic development, and this agricultural frontier has attracted farmers from various parts of Brazil due to low land prices and a topography favorable to mechanization [5,6]. Motivated by high soybean prices on the international market, local farmers are cultivating soybeans in the first year after converting the native Cerrado into agricultural areas, resulting in low yield due to the low population of nitrogen-fixing bacteria in soils, low concentrations of soil nutrients, and incomplete lime reaction in the correction of soil acidity, preventing the proper development of the crop [6]. There is a predominance of Oxisols, Argisols, and Plintosols in the region, which have a sandy and loamy/sandy texture, low buffering capacity, and low contents of clay and organic matter, resulting in low capacity for nutrient retention and medium to high acidity [7,8].

Soil acidity is a major limitation to crop productivity. The main reason for reduced yield in acidic soils is the toxicity caused by excess hydrogen (H) and aluminum (Al) [9], low saturation of bases, such as calcium (Ca), magnesium (Mg), and exchangeable potassium (K), and low availability of phosphorus (P) [10]. Despite these limitations, intensive soil management aiming for high yields has allowed for profitable yields in the first year of cultivation [6,11]. The techniques used include (i) the use of high lime rates ($CaCO_3$), which can exceed 10 to 11 t ha$^{-1}$ [8], values beyond technical concepts and recommendations contained in soil correction and fertilization bulletins used in the region [12,13]; and (ii) the use of agricultural gypsum ($CaSO_4 \cdot 2H_2O$) to improve fertility at greater depth in the soil profile.

Generally, the removal of native Cerrado vegetation and soil preparation are carried out during the dry season. Nonetheless, lime reaction effectively begins with the beginning of the rainy season, coinciding with the recommended soybean sowing period for greater yield potential. Therefore, there is not enough time for lime to completely react before plant establishment, which has been used as a justification for high lime rates in new areas for more demanding crops [8].

In the presence of water, lime dissolves and carbonate dissociates. The resulting products react with soil colloids, thereby increasing pH values, Ca and Mg contents, and base saturation. The reaction also reduces exchangeable Al, manganese (Mn) content, and phosphorus (P) fixation, thus increasing P availability for plants [14,15]. However, high lime rates can create unbalances in the soil solution because excessively high levels of Ca and Mg generate competitive inhibition between nutrients, especially potassium (K). Excess Ca can also impair Mg uptake because Ca competes with Mg for adsorption sites [16,17].

Additionally, the low solubility of lime limits the speed of reactions in Cerrado soils. Therefore, the use of agricultural gypsum is recommended due to its solubility and mobility across the soil profile [18]. Agricultural gypsum promotes the displacement of exchangeable bases to subsurface layers, reducing Al toxicity by inducing the formation of ion pairs [19], thus neutralizing toxic Al [20] and increasing Ca concentrations in deeper layers [21]. Therefore, gypsum can increase both the volume of explored soil by plant roots and plant resilience in periods of abiotic stress [22]. Thus, gypsum plays an important role in regions where low natural soil fertility and recurrent periods of water stress are inherent edaphoclimatic conditions [23].

The agricultural frontier of the Matopiba is marked by the short rainy season. For this region, little is known about the effect of applying different lime rates on chemical transformations in soil and its implications for nutrient balance and soybean yields. In the present study, the following hypotheses were tested: (i) liming rates above those recommended may sufficiently correct soil acidity, raise cation exchange capacity and Ca and Mg contents to adequate levels, even in conditions of short time for the desired reaction to occur, in areas for first-crop soybean cultivation in Matopiba; (ii) the combined

use of gypsum and lime on areas with native vegetation converted into first-crop soybean cultivation can accelerate the correction of Al toxicity in soil and minimize the need for high lime rates. The objective of this study was to evaluate the effects of high lime and gypsum applications to areas of recent conversion of native Cerrado vegetation into agriculture on soil fertility, crop nutritional status, and soybean yields in the Cerrado of Piauí.

## 2. Materials and Methods

### 2.1. Description of the Experimental Area

The experiment was carried out at União Farm, in the municipality of Currais, Piauí, Brazil (8°39′25.88″ S; 44°39′47.20″ W; 540 m altitude) (Figure S1). The region is dominated by the Cerrado biome, with an Aw climate according to the Köppen classification [24], characterized by dry winters from May to September and rainy summers from October to April. Figure 1 shows the average rainfall obtained from a rain gauge installed close to the experimental area.

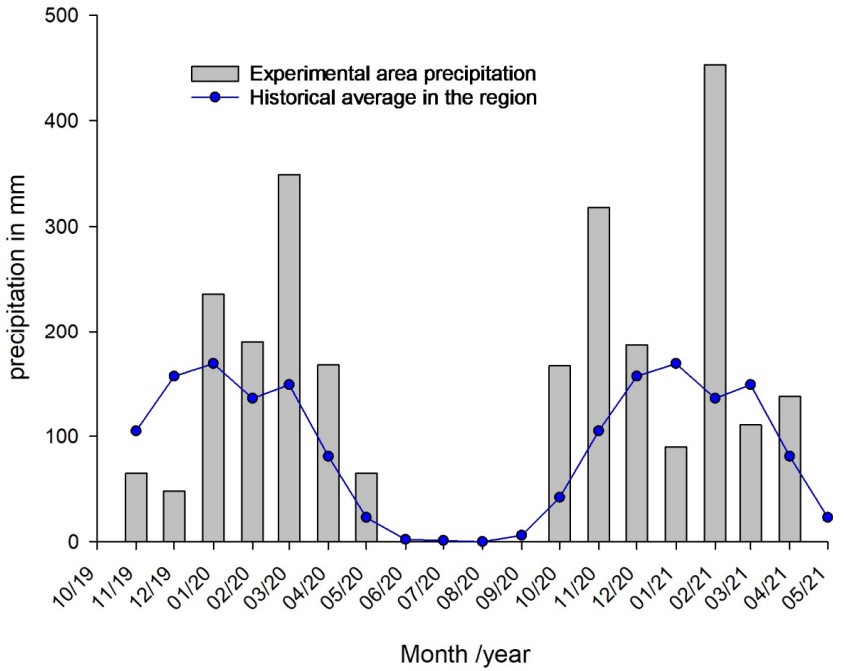

**Figure 1.** Monthly rainfall from October 2019 to May 2021 in the experimental area. Currais, Piauí (União Farm).

The soil in the experimental area is a typical dystrophic yellow Latosol (Oxisol), according to the Brazilian soil classification system [25], or haplic Xanthic Ferralsol according to the World Reference Base [26]. The study area was dominated by stricto sensu Cerrado vegetation [27]. After removing native vegetation and before installing the experiment, stratified samples were collected at depths of 0.0–0.2, 0.2–0.4, and 0.4–0.6 m, to characterize soil chemical attributes and particle size (Table 1).

To determine soil buffer capacity, the incubation method was used. Increasing amounts of limestone equivalent to doses of 0, 3, 6, and 9 t ha$^{-1}$ of dolomitic limestone (25.9% CaO, 18.7% MgO, with $CaCO_3$ equivalent of 92.7% and relative efficiency factor of 100% by granulometry < 0.3 mm) were added to the soil. Four replications were used per amount of limestone, totaling 16 soil samples. Each sample consisted of 500 g of soil stored in plastic cups and maintained soil moisture corresponding to 80% of the water retention capacity, closed and incubated for 90 days. After this period, the pH values in water were determined in a soil/solution ratio 1:2.5. The relationship pH in water × amount of $CaCO_3$ was determined using a linear equation y = 4.25 + 0.32** $R^2$ = 0.94 (*p*-value < 0.01). Soil buffering capacity expressed as the reciprocal of the slope of the curve was equal to

3.12 t/ha limestone per pH unit in the pH range of 4–7 in the study soil (at 0–0.2 m depth layer) [28].

**Table 1.** Chemical and granulometric attributes determined before the installation of the experimental area at depths of 0–0.2, 0.2–0.4, and 0.4–0.6 m in the Cerrado area.

| Layers | pH CaCl$_2$ | pH H$_2$O | TOC | N | C/N | P | K$^+$ | Ca$^{2+}$ | Mg$^{2+}$ | Al$^{3+}$ | H + Al | BC | CEC | V | m |
|---|---|---|---|---|---|---|---|---|---|---|---|---|---|---|---|
| m | | | g kg$^{-1}$ | | | mg kg$^{-1}$ | | | | % | | | | | |
| 0–0.2 | 3.76 | 4.45 | 7.94 | 0.45 | 17.6 | 1.56 | 0.02 | 0.41 | 0.12 | 0.96 | 3.37 | 0.40 | 3.91 | 13.8 | 64.1 |
| 0.2–0.4 | 3.89 | 4.48 | 5.16 | 0.39 | 13.2 | 0.65 | 0.00 | 0.01 | 0.00 | 0.56 | 2.05 | 0.01 | 2.07 | 0.3 | 98.8 |
| 0.4–0.6 | 3.98 | 4.55 | 4.17 | 0.42 | 9.9 | 0.62 | 0.00 | 0.01 | 0.02 | 0.47 | 1.50 | 0.02 | 1.52 | 1.6 | 95.2 |

| | B | | Cu | | | Fe | Mn | Zn | S-SO$_4^{2-}$ | Coarse sand | Fine sand | Total sand | Clay | Silt |
|---|---|---|---|---|---|---|---|---|---|---|---|---|---|---|
| | mg kg$^{-1}$ | | | | | | | | | | | % | | |
| 0–0.2 | 0.18 | | 0.01 | | | 66.41 | 0.02 | 0.31 | 0.79 | 44.2 | 39.4 | 83.6 | 15.5 | 0.9 |
| 0.2–0.4 | 0.15 | | 0.01 | | | 59.67 | 0.18 | 0.16 | 1.24 | 41.9 | 39.5 | 81.4 | 17.1 | 1.5 |
| 0.4–0.6 | 0.15 | | 0.02 | | | 34.35 | 0.01 | 0.46 | 2.29 | 39.7 | 36.2 | 75.9 | 22.3 | 1.8 |

### 2.2. Experimental Design, Treatments, and Management

The experiment was conducted in a randomized block design, with four replications. Five lime rates and four gypsum rates were tested, in plots measuring 13.2 × 6.6 m. The treatments consisted of the following lime rates: 0, 5, 10, 15, and 20 t ha$^{-1}$; and gypsum rates: 0, 1, 2, and 4 t ha$^{-1}$. Rates of 5 t ha$^{-1}$ and 1 t ha$^{-1}$ of lime and gypsum, respectively, corresponded to the respective recommended standard rates based on soil analysis. The standard rate was determined using a method based on eliminating Al and raising Ca + Mg [13]. The lime had the following characteristics: 25.9% CaO, 18.7% MgO, and an effective calcium carbonate equivalent (ECCE) of 77.6%. The agricultural gypsum was ground gypsum containing 15% S and 18% Ca.

After removing native Cerrado vegetation, harrowing was carried out using a 36″ grid, followed by root collection, and then another harrowing with a 36″ grid. After these operations, lime was applied over the whole area of each plot and incorporated with a 36″ harrow, followed by a leveling harrow (11 July 2019). After this operation, gypsum was broadcasted to the plots and incorporated with a 28″ harrow, followed by leveling harrowing (11 August 2019). On 2 December 2019, 880 kg ha$^{-1}$ of single superphosphate (18% P$_2$O$_5$, 15% Ca, and 8% S) were applied to all plots. On 30 December 2019, soybeans seeds (BRS9180 cultivar) were sown (2019/2020 crop year) at a spacing of 0.5 m between rows and 12.5 seeds per linear meter. At the time of sowing, 110 kg ha$^{-1}$ of MAP (50% P$_2$O$_5$, and 10% N) were also applied. 28 days after sowing (28 January 2020), 170 kg ha$^{-1}$ of KCl (62% K$_2$O) was topdressed. In the 2020/2021 crop year, sowing was carried out on 18 December 2020 using the same soybean cultivar and spacing. Base fertilization was applied using 200 kg ha$^{-1}$ of monoammonium phosphate (MAP) and sulfurgran$^®$ (78% S and 3% B) was broadcasted right before sowing. Furthermore, 30 days after sowing, 160 kg ha$^{-1}$ of KCl was applied.

In the 2019/2020 and 2020/2021 crop years, soybean seeds were inoculated with *Bradyrhizobium japonicum* at three inoculant rates. At that time, Cu, Mo, and Zn were also applied to the seeds. Throughout the development cycle, plants also received a foliar application of Mn and B at stage V4 (third completely expanded trifoliate leaf), Cu at the V6 stage (fifth completely expanded trifoliate leaf), and Mn, Cu, and B at the R1 stage (beginning of flowering).

### 2.3. Attributes Analyzed

In each plot, soil samples were collected to evaluate chemical attributes. Soil sampling was carried out by collecting four simple samples to form a composite sample, always after soybean harvest, in the 2019/2020 and 2020/2021 crop years. Samples were collected in the 0.0–0.2, 0.2–0.4 and 0.4–0.6 m layers. After sampling, the soil was air-dried and sieved through a 2 mm mesh sieve to carry out subsequent analyses. Soil chemical attributes

were: pH in $CaCl_2$ (1:2.5 soil-solution); organic carbon was quantified by wet digestion with potassium dichromate; K, P, Cu, Fe, Mn, Zn, and Na extracted with Melich-1 ($H_2SO_4$ 0.0125 mol $L^{-1}$ and HCl 0.050 mol $L^{-1}$). K and Na concentrations were determined by flame photometer, P by colorimetry; Ca, Mg and Al were extracted with potassium chloride (1 mol $L^{-1}$) and determined by atomic absorption, together with Cu, Fe, Mn and Zn; $Al^{3+}$ was determined by titration; potential acidity (H + Al) was extracted with calcium acetate solution (0.5 mol $L^{-1}$) and determined by titration; S was determined through the turbidity formed by sulfate, using barium chloride [29,30].

Specifically for samples at a depth of 0.0–0.2 m, the concentrations of non-exchangeable Ca and Mg in soil were also determined, which represent the fraction of the amendments not yet reacted. These analyses were carried out according to the method described by [31–34]. Briefly, in a percolator tube, 5 $cm^3$ of soil was added to a filter paper, and 50 mL of the KCl solution was subsequently percolated through the sample (first extraction). In a 1 mL aliquot of the extract obtained after percolation, 10 mL of 0.1% lanthanum (La) was added, and Ca and Mg contents were measured using atomic absorption spectrophotometry. After the first extraction, the soil was transferred to an Erlenmeyer flask, and 15 mL of $H_2O$ + 25 mL of 0.8 mol $L^{-1}$ HCl were added. The solution was boiled on a hot plate for 5 min (200 °C). Therefore, by boiling the soil together with HCl, the remaining lime is forced to react (rapid dissolution). Then, after cooling, the material was filtered (second extraction). In 1 mL of the extract, 10 mL of La (0.5%) was added. The reading was carried out using spectrophotometry, determining the Ca and Mg contents, referring to estimated values of non-exchangeable Ca and Mg. The results, after discounting the values of the control, which did not receive lime and gypsum, indicate the fraction of the amendments that had not yet reacted in soil.

To evaluate the nutritional status of soybeans, recently expanded leaves with petiole, corresponding to the 3rd and 4th trifoliate leaf from the main stem, were collected in the two crop years, in the period between the beginning of flowering (R1) and full flowering (R2) [35]. Leaf macro- and micronutrient contents (P, K, Ca, Mg, S, Cu, Fe, Mn, and Zn) were quantified according to [36]. After nitric and perchloric digestion, P was determined by colorimetry, K, Ca, Mg, Cu, Fe, Mn, and Zn by atomic absorption, S by turbidimetry, N by the Kjeldahl method, and B by calcination. Soybean yields were determined by manually harvesting two 2 m long rows in the center of the plot, with subsequent standardization of seed moisture to 13% and conversion of grain yield to kg $ha^{-1}$.

### 2.4. Data Analysis

The data were subjected to the normality test (Shapiro–Wilk, $p < 0.05$), with subsequent analysis of variance using the F test ($p < 0.05$). When a significant effect was found for the lime factor, gypsum, and/or their interaction, the data means were analyzed using a regression test on the statistical software SISVAR version 5.6 [37].

## 3. Results

### 3.1. Soil Fertility in the First Growing Season

In the 2019/2020 crop year, there was a significant interaction between lime and gypsum rates for P, K, m, and S-$SO_4^{2-}$ in the 0.0–0.2 m layer. In isolation, lime rates resulted in significant differences for the variables pH, P, K, Ca, Mg, Al, H + Al, BC, BS, m, S-$SO_4^{2-}$, Fe, Mn, and Zn. Likewise, gypsum rates promoted significant changes for Ca, BC, m, and S-$SO_4^{2-}$ (Table S1). The changes promoted by lime rates in the 0.0–0.20 layer allowed fitting quadratic models for pH (Figure 2a), Ca (Figure 2e), Mg (Figure 2f), H + Al (Figure 2h), BC (Figure 2i), BS% (Figure 3b), and Fe (Figure S2). For Al contents, a decreasing exponential model was fitted (Figure 2g). For P, quadratic models were fitted for the lime rate at all gypsum rates (Figure 2b). For K, quadratic models were fitted only to the lowest gypsum rate (Figure 2c), and for S-$SO_4^{2-}$, at the highest and lowest gypsum rates (Figure 3e). The interaction between lime and gypsum for Al saturation (m) resulted in fitting exponential models as a function of lime rates at all gypsum rates (Figure 3c). For

Mn and Zn (Figure S2), decreasing linear models were fitted to lime rates. The application of lime at rates of 20, 10.8, 10.5, 20, 20, 20, 20, and 10.4 t ha$^{-1}$ resulted in the highest values of pH$_{CaCl_2}$ (5.55), pH$_{H_2O}$ (6.11) P (21 mg kg$^{-1}$), K (42.96 mg kg$^{-1}$), Ca (1.41 cmol$_c$ kg$^{-1}$), Mg (0.86 cmol$_c$ kg$^{-1}$), BC (2.34 cmol$_c$ kg$^{-1}$), BS (63.8%) and S-SO$_4$$^{2-}$ (8.59 mg kg$^{-1}$). In turn, lime rates of 15, 20, 15, 20, 20, and 20 t ha$^{-1}$ resulted in the lowest values of Al (zero cmol$_c$ kg$^{-1}$), H + Al (1.4 cmol$_c$ kg$^{-1}$), m (zero cmol$_c$ kg$^{-1}$), Fe (41.41 mg kg$^{-1}$), Mn (24.84 mg kg$^{-1}$), and Zn (1.61 mg kg$^{-1}$).

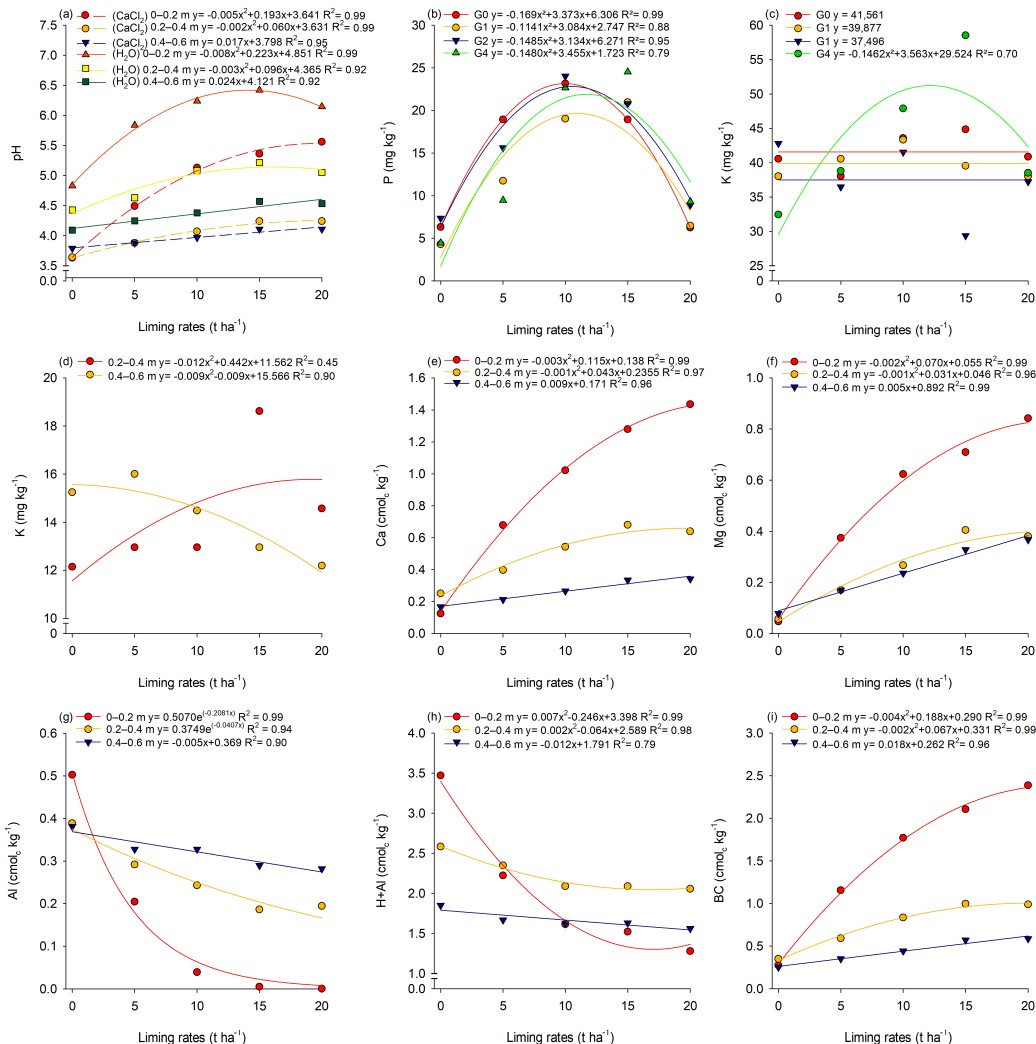

**Figure 2.** Soil pH values as a function of lime rates (**a**); P (**b**) and K (**c**) as a function of the interaction between lime and gypsum rates at a depth of 0.0–0.20 m; K as a function of lime rates at depths of 0.20–0.40 and 0.40–0.60 m (**d**); Ca (**e**), Mg (**f**), Al (**g**), H + Al (**h**) and BC (**i**) as a function of lime rates in the 2019/2020 crop year. Currais, Piauí, Brazil.

The highest gypsum rate (4 t ha$^{-1}$) provided higher Ca concentrations (1.8 cmol$_c$ kg$^{-1}$), BC (2.5 cmol$_c$ kg$^{-1}$), and S-SO$_4$$^{2-}$ (5.78 mg kg$^{-1}$) and the lowest concentration of Al (0.28 cmol$_c$ kg$^{-1}$). Regardless of gypsum rate, the lime rates provided maximum P concentration ranging from 10 and 11.7 t ha$^{-1}$, with values of 22.34, 18.98, 22.03, and 21.08 mg kg$^{-1}$ of P at rates of 0, 1, 2, and 4 t ha$^{-1}$ of gypsum, respectively. The rate of 4 t ha$^{-1}$ of gypsum combined with 12.1 t ha$^{-1}$ of lime led to the highest K concentration (51.17 mg kg$^{-1}$). Aluminum saturation reduced to virtually zero at rates of 15.7, 15.9, 15.7, and 15.6 t ha$^{-1}$ of lime combined with 0, 1, 2, and 4 t ha$^{-1}$ of gypsum, respectively. The lime rates of 8.4 and 9.5 t ha$^{-1}$ combined with 0 and 4 t ha$^{-1}$ of gypsum provided the highest concentrations of S-SO$_4$$^{2-}$ (8.28 and 10.94 mg kg$^{-1}$, respectively).

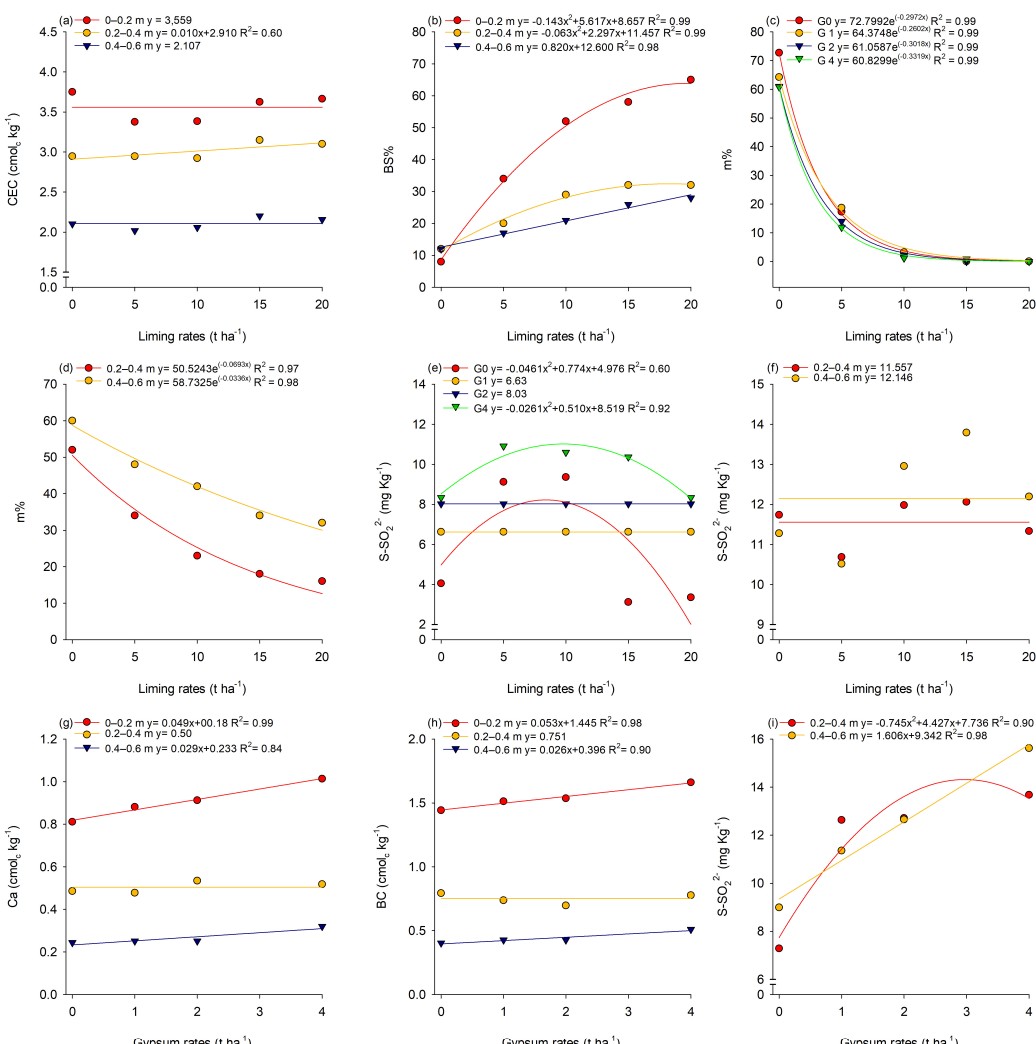

**Figure 3.** Soil CEC values at a depth of 0.20–0.40 m (**a**) and BS% (**b**) as a function of lime rates, aluminum saturation—m (**c**) as a function of the interaction between lime and gypsum rates at a depth of 0.0–0.20 m, aluminum saturation—m (**d**) as a function of lime rates at depths of 0.20–0.40 and 0.40–0.60 m, S-SO$_4^{2-}$ (**e**) as a function of the interaction between lime and gypsum rates at a depth of 0.0–0.20 m, S-SO$_4^{2-}$ (**f**) as a function of lime rates at depths of 0.20–0.40 and 0.40–0.60 m, Ca (**g**), BC (**h**), and S-SO$_4^{2-}$ (**i**) as a function of gypsum rates, in the 2019/2020 crop year. Currais, Piauí, Brazil.

In the 0.2–0.4 m layer, there was no significant interaction between lime and gypsum rates (Table S2). However, lime rates influenced pH, K, Ca, Mg, Al, H + Al, BC, CEC, BS and m values. Similarly, gypsum rates significantly affected Ca, BC, and S-SO$_4^{2-}$ values. Regarding lime rates, quadratic models were fitted for pH (Figure 2a), K (Figure 2d), Ca (Figure 2e), Mg (Figure 2f), H + Al (Figure 2h), BC (Figure 2i), and BS (Figure 3b); exponential models for Al (Figure 2g); linear model for CEC (Figure 3a) and Fe (Figure S2); and exponential model for m (Figure 3d). The highest pH$_{H_2O}$ was obtained with 14 t ha$^{-1}$ of lime. However, the highest liming rate resulted in the highest values of pH$_{CaCl_2}$ (4.8), K (15.72 mg kg$^{-1}$), Ca (0.6 cmol$_c$ kg$^{-1}$), Mg (0.4 cmol$_c$ kg$^{-1}$), BC (1.61 cmol$_c$ kg$^{-1}$) and BS (35%). The highest lime rate also led to the lowest values of Al (0.16 cmol$_c$ kg$^{-1}$), H + Al (2.02 cmol$_c$ kg$^{-1}$), m (11%), and Fe (136.29 mg Kg$^{-1}$). As for the gypsum rates, a quadratic model was fitted to the values of S-SO$_4^{2-}$ (Figure 3i), with the rate of 3.0 t ha$^{-1}$ resulting in a higher S-SO$_4^{2-}$ concentration (14.27 mg kg$^{-1}$).

For the 0.4–0.6 m layer, there was no significant effect of the interaction between lime and gypsum rates (Table S3). However, lime rates significantly influenced the values of pH

(Figure 2a), K (Figure 2d), Ca (Figure 2e), Mg (Figure 2f), Al (Figure 2g), H + Al (Figure 2h), BC (Figure 2i), BS (Figure 3b), and m (Figure 3d). Similarly, gypsum rates altered the values of Ca (Figure 3g), BC (Figure 3h), S-SO$_4{}^{2-}$ (Figure 3i), BS, and m (Figure S2). Overall, there was a linear model fit to the data for all variables, except aluminum saturation (m), for which a quadratic model was fit. At this depth, the highest lime rate (20 t ha$^{-1}$) promoted the highest values of pH$_{CaCl_2}$ (4.15), pH$_{H_2O}$ (4.60), Ca (0.36 cmol$_c$ kg$^{-1}$), Mg (0.22 cmol$_c$ kg$^{-1}$), BC (0.62 cmol$_c$ kg$^{-1}$), and BS (29%). Likewise, this lime rate resulted in the lowest values of K (12.21 mg kg$^{-1}$), Al (0.27 cmol$_c$ kg$^{-1}$), H + Al (1.53 cmol$_c$ kg$^{-1}$), and m (32%). As for gypsum, the highest rate (4 t ha$^{-1}$) led to the highest values of Ca (0.31 cmol$_c$ kg$^{-1}$), BC (0.50 cmol$_c$ kg$^{-1}$), BS (22%), and S-SO$_4{}^{2-}$ (15.80 mg kg$^{-1}$), and the lowest value of m (37%).

### 3.2. Soil Fertility in the Second Growing Season

In the 2020/2021 crop year, there was no significant interaction between the rates of lime and gypsum, for any of the attributes analyzed in the 0–0.2 m layer. (Table S4). However, there was a significant effect of lime rates on the values of pH (Figure 4a), P (Figure 4b), K (Figure 4c), Ca (Figure 4d), Mg (Figure 4e), Al (Figure 4f), H + Al (Figure 4g), BC (Figure 4h), CEC (Figure 4i), BS (Figure 5a), m (Figure 5b), S-SO$_4{}^{2-}$ (Figure 5c), and Fe (Figure S2). In general, quadratic models were fitted to all variables, except for Al and Fe, for which the fitted models were exponential and linear, respectively. The lime rates of 20, 15.5, 17.2, 17.6, 18.1, 17.6, 11.9, 20.0, and 12.3 t ha$^{-1}$ provided higher pH values (5.62 $_{CaCl_2}$, 6.52 $_{H_2O}$), P (10.39 mg kg$^{-1}$), K (40.86 mg kg$^{-1}$), Ca (1.56 cmol$_c$ kg$^{-1}$), Mg (0.86 cmol$_c$ kg$^{-1}$), BC (3.52 cmol$_c$ kg$^{-1}$), CEC (4.76 cmol$_c$ kg$^{-1}$), BS (76%) and S-SO$_4{}^{2-}$ (3.67 mg kg$^{-1}$), respectively. In turn, lime rates of 16.3, 20, 15.9, and 20 t ha$^{-1}$ resulted in the lowest values of Al (0 cmol$_c$ kg$^{-1}$), H + Al (1.09 cmol$_c$ kg$^{-1}$), m (0%) and Fe (46.25 mg kg$^{-1}$), respectively. In this crop year, there was no effect of gypsum rates on the variables.

At the depth of 0.2–0.4 m, there was no significant interaction between lime and gypsum rates (Table S5). A significant isolated effect of lime rates was observed for the values of pH (Figure 4a), Ca (Figure 4d), Mg (Figure 4e), Al (Figure 4f), H + Al (Figure 4g), BC (Figure 4h), CEC (Figure 4i), BS (Figure 5a), m (Figure 5b), S-SO$_4{}^{2-}$ (Figure 5c), and Fe (Figure S2). As for gypsum rates, there was a significant effect only for the S-SO$_4{}^{2-}$ concentration (Figure 5d). The highest lime rate (20 t ha$^{-1}$) provided the highest values of pH$_{CaCl_2}$ (4.7), pH$_{H_2O}$ (4.9), Ca (1.18 cmol$_c$ kg$^{-1}$), Mg (0.82 cmol$_c$ kg$^{-1}$), BC (1.24 cmol$_c$ kg$^{-1}$), CEC (3.52 cmol$_c$ kg$^{-1}$), and BS (55%) and at the lowest values of Al (0.02 cmol$_c$ kg$^{-1}$), H + Al (1.56 cmol$_c$ kg$^{-1}$), S-SO$_4{}^{2-}$ (4.18 mg kg$^{-1}$), and Fe (115.32 mg kg$^{-1}$). The application of 2.2 t ha$^{-1}$ of gypsum resulted in the highest S-SO$_4{}^{2-}$ concentration (6.37 mg kg$^{-1}$).

For the depth of 0.4–0.6 m, there was no significant interaction between lime and gypsum rates (Table S6). Lime rates significantly influenced the values pH (Figure 4a), K (Figure 4c), Ca (Figure 4d), Mg (Figure 4e), Al (Figure 4f), H + Al (Figure 4g), BC (Figure 4h), CEC (Figure 4i), BS (Figure 5a) in m (Figure 5b), with quadratic models fitted to all variables, except m, to which a decreasing exponential model was fitted. Gypsum rates significantly altered the values of S-SO$_4{}^{2-}$ (Figure 5d) and Cu (Figure S2), to which increasing and decreasing linear models were fitted, respectively. The application of the highest lime rate (20 t ha$^{-1}$) resulted in the highest values of pH$_{CaCl_2}$ (4.45), pH$_{H_2O}$ (4.6), Ca (0.69 cmol$_c$ kg$^{-1}$), Mg (0.34 cmol$_c$ kg$^{-1}$), BC (1.15 cmol$_c$ kg$^{-1}$), CEC (2.58 cmol$_c$ kg$^{-1}$), and BS (44%), and at the lowest values of Al (0.14 cmol$_c$ kg$^{-1}$), H + Al (1.43 cmol$_c$ kg$^{-1}$) in m (14%). The gypsum rate of 2.9 t ha$^{-1}$ resulted in the lowest value of S-SO$_4{}^{2-}$ (12.06 mg kg$^{-1}$) and the rate of 4.0 t ha$^{-1}$ of gypsum provided the lowest value of Cu (0.17 mg kg$^{-1}$).

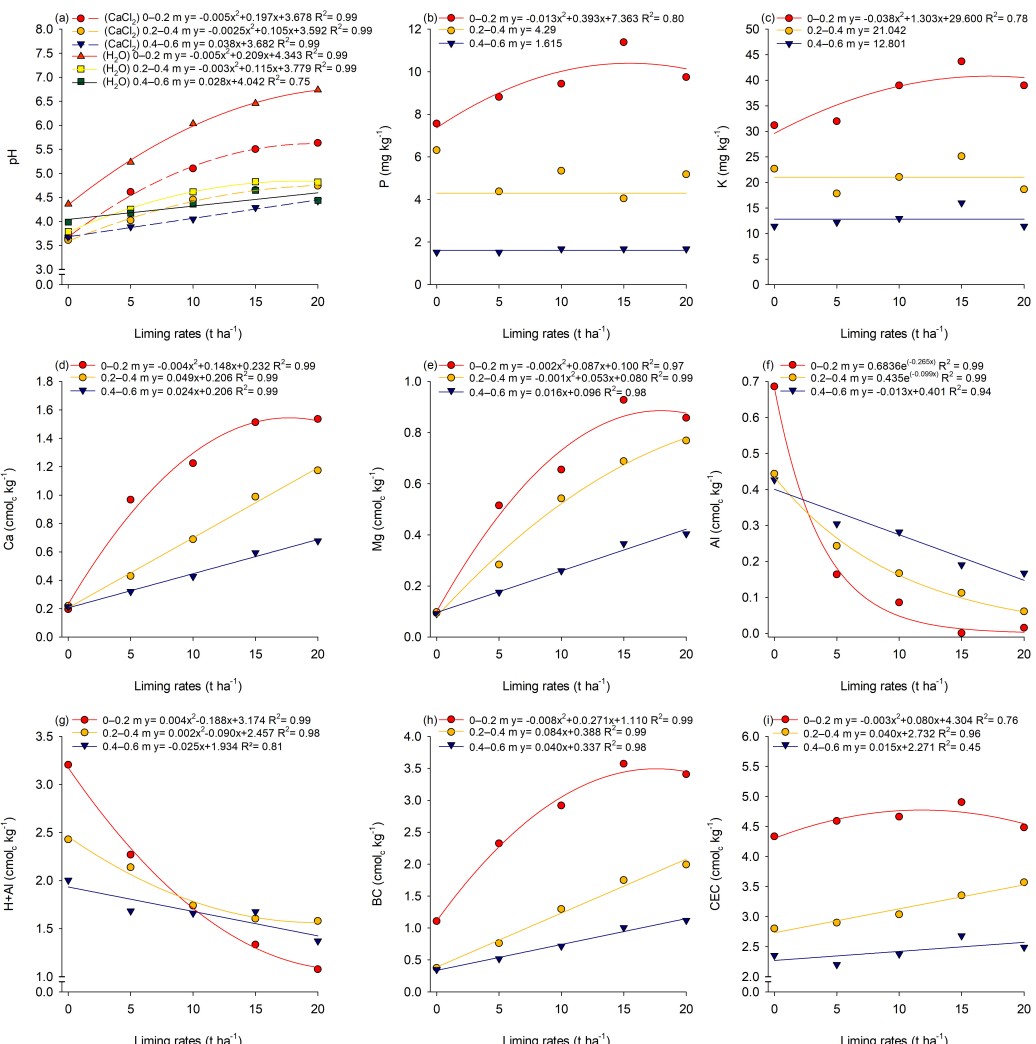

**Figure 4.** Data on pH (**a**), P (**b**), K (**c**), Ca (**d**), Mg (**e**), Al (**f**), H + Al (**g**), BC (**h**), and CEC (**i**) as a function of lime rates at depths of 0.0–0.10, 0.20–0.40, and 0.40–0.60 m in the 2020/2021 crop year. Currais, Piauí, Brazil.

### 3.3. Non-Exchangeable Calcium and Magnesium in the First and Second Growing Season

In the 2019/2020 crop year, the concentrations of non-exchangeable Ca, Mg, and Ca + Mg were significantly influenced by the lime and gypsum rates individually and by the interaction of these factors, except non-exchangeable Mg (Table S7). However, in the 2020/2021 crop year, there was a significant effect only of lime rates on the values of non-exchangeable Ca, Mg, and Ca + Mg. In general, increasing linear models were fitted for all variables, except for Mg vs. lime rates in the 2019–2020 crop year, to which a quadratic model was fitted (Figure 6).

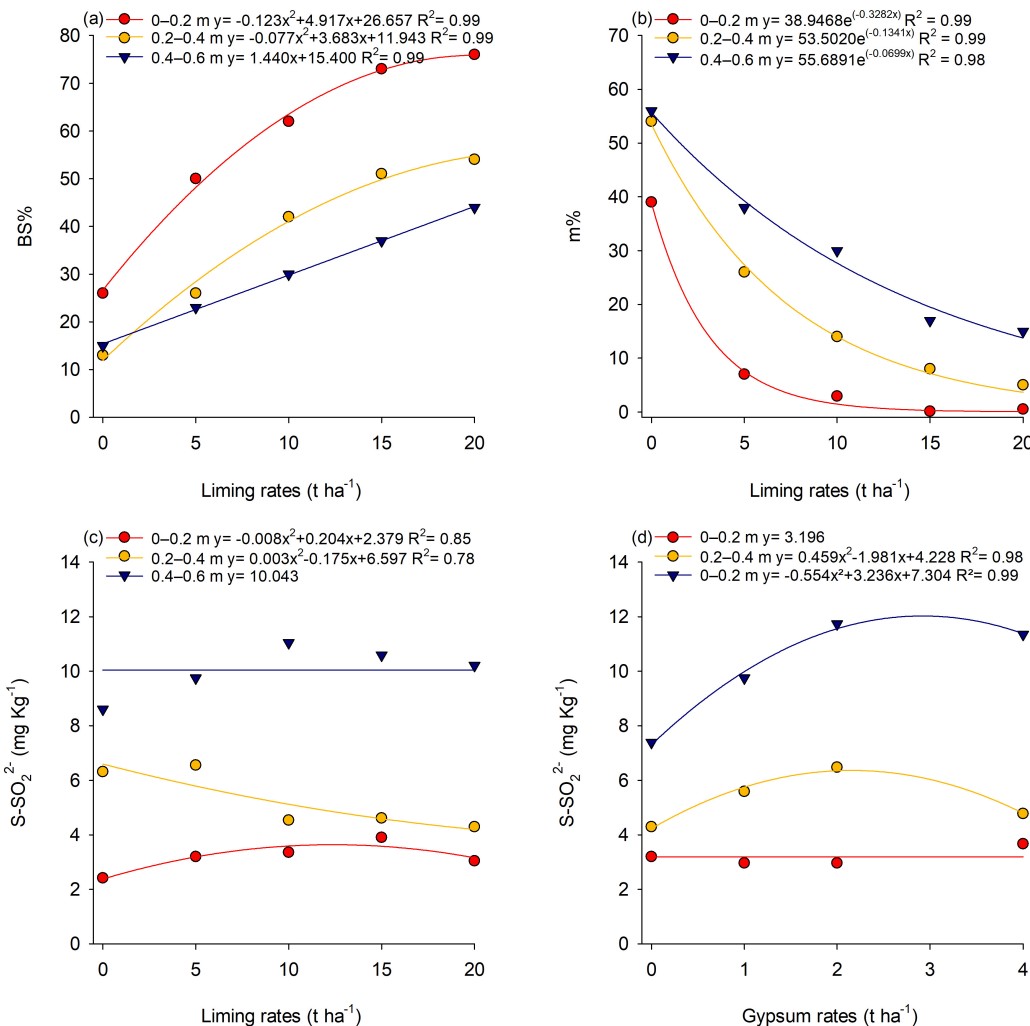

**Figure 5.** Data on BS% (**a**), aluminum saturation—m (**b**), and S-SO$_4^{2-}$ (**c**) as a function of lime rates and S-SO$_4^{2-}$ (**d**) as a function of gypsum rates at depths of 0.0–0.10, 0.20–0.40, and 0.40–0.60 m, in the 2020/2021 crop year. Currais, Piauí, Brazil.

*3.4. Nutritional Status of Soybeans in the First and Second Growing Season*

There was no interaction between lime and gypsum rates for macro and micronutrient contents in soybean leaves, in the two crop years evaluated (Table S8). In the 2019/2020 crop year, the contents of Ca and Mg (Figure 7a,c), in addition to N, P, K, Cu, Fe, Mn, and Zn (Figure S3) were influenced by lime rates. Gypsum rates influenced Ca, Mg, and S values (Figure 7b,d,f), in addition to Zn (Figure S3). In the 2020/2021 crop year, lime rates influenced Ca and Mg contents (Figure 7a,c), as well as N, P, K, Fe, Mn, and Zn (Figure S4). Gypsum rates significantly affected K and B values (Figure S4). In the 2019/2020 crop year, the highest lime rate (20 t ha$^{-1}$) reduced foliar levels of N (36 g kg$^{-1}$), P (2.5 g kg$^{-1}$), Fe (124 mg kg$^{-1}$), Mn (96 mg kg$^{-1}$), and Zn (23 mg kg$^{-1}$). However, this same rate resulted in higher concentrations of K (22 g kg$^{-1}$), Ca (8.5 g kg$^{-1}$), Mg (4.9 g kg$^{-1}$) and Cu (3.3 mg kg$^{-1}$). Gypsum rates of 2.3, 1.2, 4.0, and 4.0 t ha$^{-1}$ provided the highest concentrations of Ca, Mg, S and Zn, respectively. In the 2020/2021 crop year, lime rates of 10.4, 20, 11.5, 20, and 16.6 t ha$^{-1}$ provided the highest levels of N (39.9 g kg$^{-1}$), P (4.4 g kg$^{-1}$), K (24.2 g kg$^{-1}$), Ca (6.9 g kg$^{-1}$), and Mg (6 g kg$^{-1}$). However, the highest lime rate (20 t ha$^{-1}$) led to the lowest levels of Fe (59 mg kg$^{-1}$) and Mn (13 mg kg$^{-1}$). As for gypsum, the highest rate (4 t ha$^{-1}$) resulted in the lowest concentrations of K (22.5 g kg$^{-1}$) and B (40 mg kg$^{-1}$) (Figure 7).

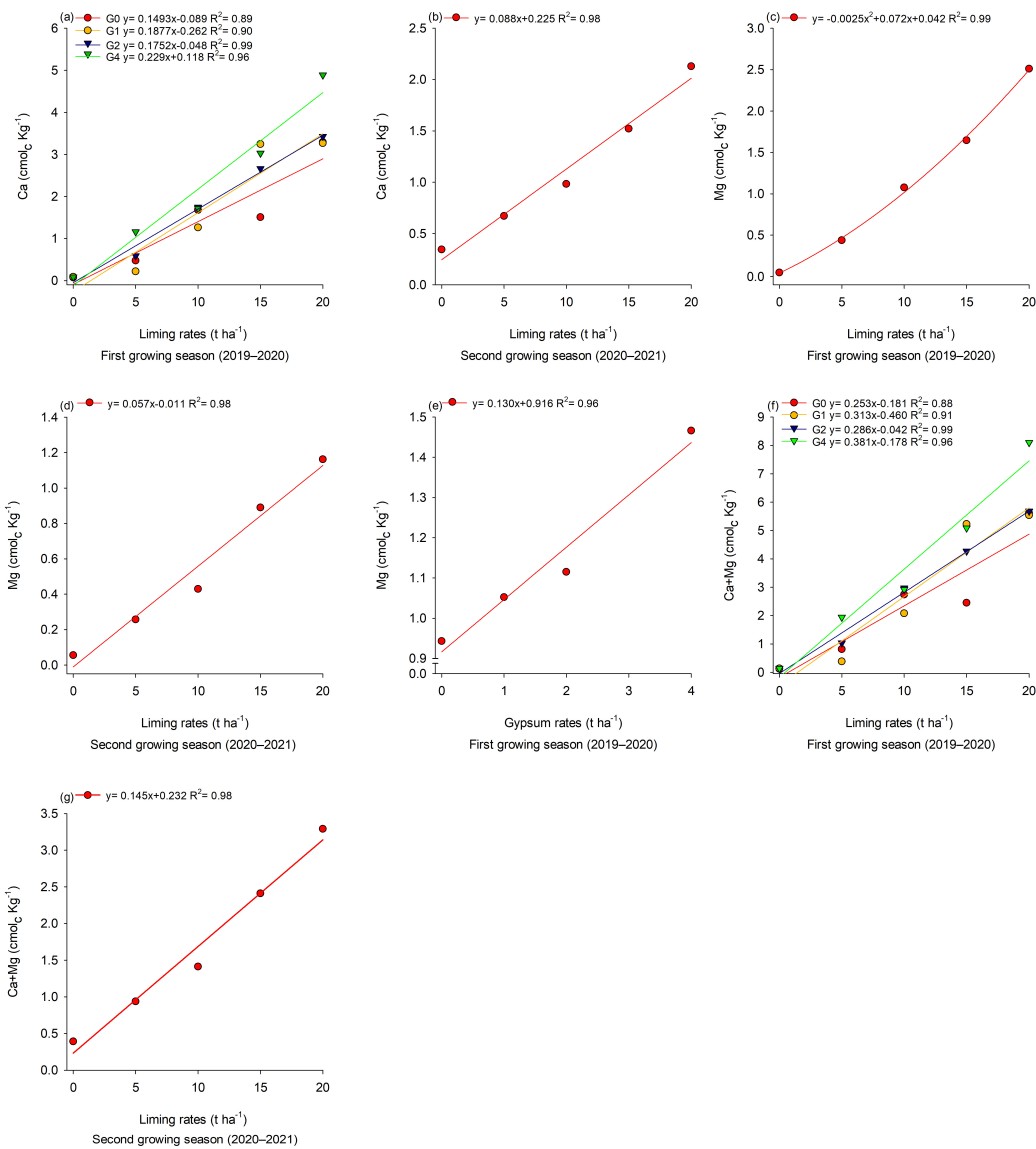

**Figure 6.** Non-exchangeable Ca (**a**,**b**), non-exchangeable Mg (**c**,**d**), non-exchangeable Ca + Mg as a function of lime rates (**f**,**g**), and non-exchangeable Mg as a function of gypsum rates (**e**) in the 0–0.2 m layer of soil, in the 2019/2020 and 2020/2021 harvests. Currais, Piauí.

### 3.5. Soybean Yield in the First and Second Growing Season

There was no significant effect of gypsum rates and their interaction with lime on soybean grain yield in the two crop years. However, a significant effect of lime rates on grain yield was observed in both crop years (Table S9), to which sigmoidal models were fitted in order to explain the biological behavior of the data. These models showed a tendency for yield to stabilize passed the lime rate of 10 t ha$^{-1}$ (Figure 8).

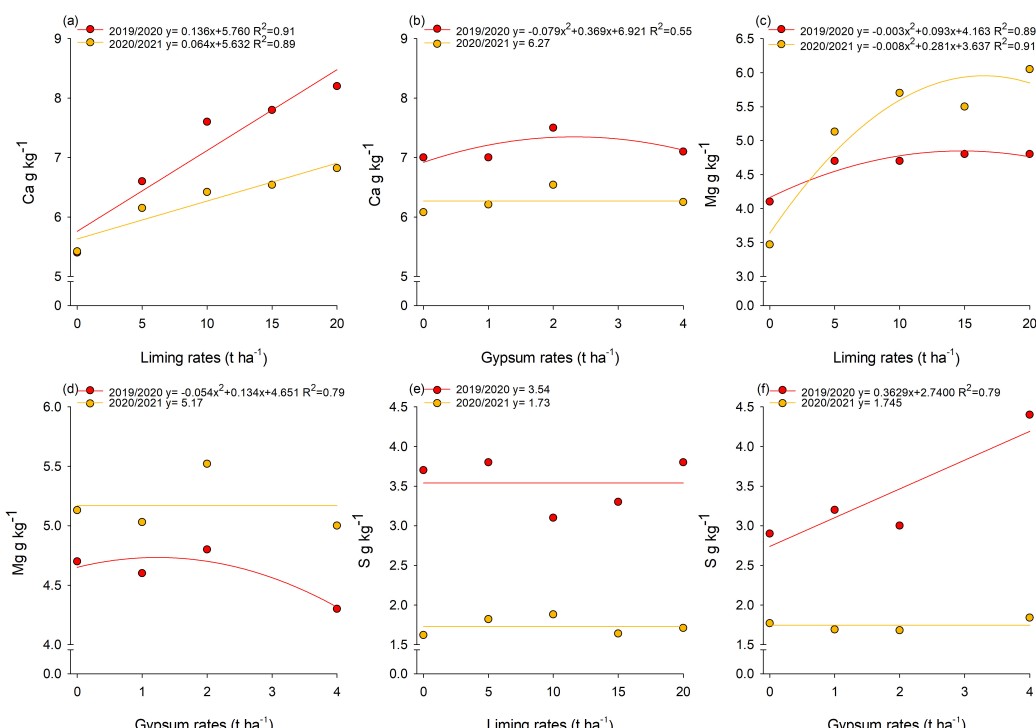

**Figure 7.** Leaf contents of Ca (**a**), Mg (**c**) and S (**e**) as a function of lime rates, Ca (**b**), Mg (**d**) and S (**f**) as a function of gypsum rates in the 2019/2020 and 2020/2021 harvests. Currais, Piauí.

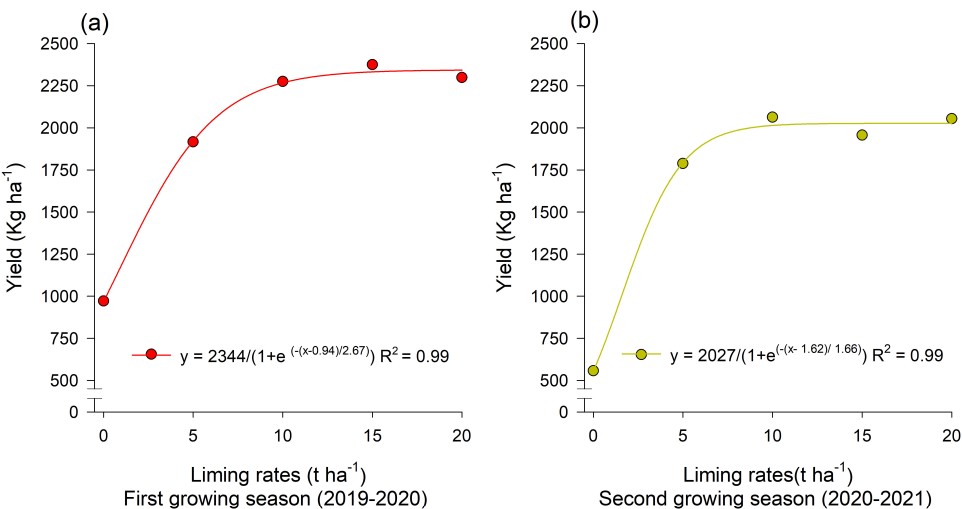

**Figure 8.** Soybean grain yield as a function of lime rates in the 2019/2020 (**a**) e 2020/2021 (**b**) crop years. Currais, Piauí.

## 4. Discussion

Liming using rates higher than those recommended by official soil correction and fertilization recommendation manuals for the Cerrado [13] has led to a rapid improvement in soil chemical properties in the three layers analyzed. Chemical indicators associated with the soil correction process increased in the 0.0–0.2 m layer (Ca, Mg, BC, and SB) when using the highest lime rate (20 t ha$^{-1}$), while other indicators were reduced (Al, H + Al, m, and cationic micronutrients) from the first year after application. Specifically for Al and m, lime rates close to 15 t ha$^{-1}$ practically completely neutralized the toxicity caused by Al. The increase in soil pH resulting from liming increases the concentration and activity of OH$^{-}$ ions in the soil solution, leading to Al precipitation, thereby becoming unavailable [38].

Lime rates between 10 and 15 t ha$^{-1}$ increased soil concentrations of P and K, but rates above this range (between 15 and 20 t ha$^{-1}$) tended to reduce the concentrations of these elements and cationic micronutrients. Such a reduction can lead to a nutritional deficiency in plants, with consequent limitation in grain yield [39]. This logic is corroborated by the results of the foliar diagnosis, which demonstrated a reduction in N, P, and micronutrient levels in the first harvest of soybean plants under high lime rates. Therefore, the high lime rates possibly resulted in an imbalance of P and cations. Ca and Mg are cations that interact with elements in the soil solution, competing for adsorption sites, thus precipitating cations and anions depending on their availability [40]. Liming, up to a certain limit, promotes an increase in P concentrations in soil due to a decrease in P adsorption to Fe and Al oxides [41]. However, very high lime rates raise soil pH and can favor the formation of calcium phosphates, resulting in unavailable P. This can occur even at pH below 7.0 because the residual effect of lime with the increase in soluble Ca (liming) contributes to P precipitation [42]. P adsorption has a "U" shape in relation to the pH scale. However, when the effects of pH on P availability are considered, the desorption rate as a function of pH and its effects on the rate of P absorption by plant roots must be considered [43], which helps to explain the decrease in P levels in plant tissue at the highest lime rates in the first year of lime application. Just like P, the effects of pH modulation due to lime application on nutrient uptake by plants also apply to other elements absorbed in the form of anions, such as S.

The reduction in the levels of cationic micronutrients due to the increase in pH is justified by the complexation or chelation of Cu, Fe, and Mn by humic acids (organic matter) and the transformation of Zn into non-soluble forms [44]. Furthermore, there may be a change in the soil redox potential due to the disturbance caused by lime and gypsum incorporation [41]. Cationic micronutrients (Cu, Mn, and Zn) show greater adsorption at higher pH values due to the reaction with hydroxyls. However, this effect is distinct among micronutrients, such as Zn [43]. As for B, its low response to lime and gypsum application can be associated with the way B is absorbed by plants. The applied form (boric acid) has no charge and is therefore little influenced by soil pH. Furthermore, the application of boron-based fertilizer (sulfurgran®) in the second harvest may have contributed to the absence of B deficiency [43].

For K, in turn, lime application may reduce K concentration in soil because of the displacement of K$^+$ from the Ca and Mg exchange complex, moving the ions downwards into the soil profile, so K may be lost through leaching [41]. It is believed that this mechanism was responsible for the decrease in K concentrations in the highest lime rates. With an increase in soil pH, there is an increase in negative charges on soil particles, which implies a lower proportion of cations present in the soil solution and a consequent decrease in the rate of movement to the roots by diffusion [43].

In the 0.2–0.4 m layer, there were increases for most soil chemical attributes in both harvests (pH, K, Ca, Mg, BC, CEC, and BS), in addition to a reduction in Al, H + Al, m and Fe due to lime application. Similar results were observed in the 0.4–0.6 m layer. The consistent reductions in the values of acidity components and improvement in the soil chemical properties at all depths in the two crop years suggest that after the incorporation of lime, there was a uniform liming process [45]. Lime application increasing CEC was due to the presence of pH-dependent charges, which in turn was due to the presence of the clay mineral kaolinite. The justification for this statement is that kaolinite is present during the genesis of the studied soil [25] and pH-dependent charges constitute more than 50% of the surface changes of this clay mineral [46]. Another important aspect is that the increase in CEC was observed not only in topsoil but also in deeper layers, which illustrates the importance of a good incorporation of the corrective agent to improve soil fertility in depth.

There is consensus in the literature about the positive effects of applying gypsum in acidic soils include reducing toxic Al levels in the subsurface and improving the rooting and tolerance of plants in water scarcity-prone environments [47,48]. This finding is important because the Matopiba region has a highly variable climate [49,50]. Although this region is

characterized by well-defined rainy seasons (between October and April) and dry seasons (between May and September), the climate change scenario with extreme weather events tends to alter the pattern, frequency, and distribution of weather events, thereby directly impacting agricultural production [51]. Therefore, the use of gypsum and lime proved to contribute to accelerating the construction of the soil profile, as seen in the first and second harvests. The construction of a soil profile with favorable chemical characteristics up to a depth of 0.6 m promotes conditions for a better root environment, mainly due to the increase in Ca and S contents. The association between both nutrients forms a soluble ion pair with a lower valence or neutral charge, favoring the mobility of Ca and S in depth with increasing gypsum application rates [38].

In the 2019/2020 crop year, the highest gypsum rate could mitigate P fixation and decrease K concentration in topsoil (0.0–0.2 m). Furthermore, high gypsum rates altered Mg levels in plant tissue in the first harvest and K levels in the second harvest, although this behavior was not observed for soil concentrations of the two nutrients. A possible reduction in K and Mg content in soil due to gypsum application is justified by the formation of an ionic pair of these cations with sulfate, thus decreasing the concentration of both nutrients in superficial layers [52,53]. This effect was probably mitigated by the presence of considerable Mg concentrations in the lime, and by K fertilization, which allowed the maintenance of adequate nutrient levels in the diagnostic soybean leaf [35].

However, the absence of effects of gypsum application on soybean yields was not expected, especially considering the high Al concentrations in topsoil in the present study, and the known positive effects of gypsum in reducing Al concentration [52]. It is possible that lime application at high rates, associated with a good incorporation of lime into the soil and combined with the application of single superphosphate and sulfurgran® (S sources), have minimized a possible positive effect of gypsum on soybean yield. In a systematic review of the effect of applying gypsum in a subtropical environment to a subsoil showing high acidity and water deficiency, it was observed that there was a change in soybean yield in only 23% of the cases analyzed [48]. Other studies report that soybeans are less responsive to gypsum application when compared to grasses due to greater Ca uptake and less dependence on N from the soil by soybeans [48,53].

In the 2019/2020 crop year, it was found that lime and gypsum rates influenced the values of Ca, Mg, and non-exchangeable Ca + Mg (residual lime), and the higher the rate, the higher the residual contents that can still react with soil over time in the presence of moisture. However, in the 2020/2021 crop year, only liming influenced the concentrations of non-exchangeable Ca, Mg and Ca + Mg. The effect of gypsum on the amount of residual lime in the 2019/2020 crop year can be justified by the reduction in lime dissolution [34] because topsoil (0–0.2 m) showed a higher measured pH value. The data on the amounts of residual lime, when converted to the percentage of lime that effectively reacted (Table S10), demonstrated that there was a low reaction rate in the 2019/2020 crop year when the standard rate of 5 t ha$^{-1}$ (51% residual lime) was applied. This information confirms the hypothesis that high lime rates are necessary to improve the soil environment in conditions of little time available for reactions to occur, thus allowing the cultivation of first-crop soybeans in newly open areas. It is worth highlighting that the short time available for lime reaction, which was between lime application and soybean sowing, is due to the short duration of the sowing window after the beginning of the rainy season, determined based on the agroclimatic risk zoning (ZARC) for the region [49].

Therefore, the use of lime at higher rates than those recommended in official bulletins is necessary to compensate for differences in water availability in the Matopiba region in comparison with prevailing conditions of the regions where the data used in these bulletins were generated. Furthermore, most of the fertility manuals still in use were produced based on research carried out in the 1980s and 1990s. Currently, soybean cultivars exhibit different characteristics, demanding more intensive soil management systems, such as greater supply of biological and nutritional inputs. These characteristics changed the dynamics of nutrient demand over time, justifying the need to review official recommendations, especially for

agricultural frontier regions with peculiar conditions. Proceeding analogous reasoning to that of [45], and considering the results of soil analysis in the 0–0.2 and 0–0.4 m layers, the amount of lime needed to increase base saturation to 70% [54] at the beginning of the experiment would be 3.82 t h$^{-1}$ and 5.90 t ha$^{-1}$ (lime with ECCE of 77.6%), values that are much lower than those that resulted in the best responses in the present study.

Soybean grain yield was significantly increased up to a lime rate of 10 t ha$^{-1}$, after which grain yield stabilized in both crop years. This productive increase was associated with the improvement of the soil environment in the layer where root development of soybean plants predominates [55]. The lack of increase in yields at lime rates above 10 t ha$^{-1}$ can be justified by the resulting nutritional imbalance, as seen for P and K. The data from the present study suggest that this is the most appropriate rate for newly open areas with similar characteristics to the Matopiba region and meets the quantities applied by many farmers in the region. The increase in grain yield at the rate of 10 t ha$^{-1}$ was 18% and 12% in relation to the standard rate for the 2019/2020 and 2020/2021 crop years, respectively. Furthermore, if the official lime recommendation for soybean cultivation were considered based on the base saturation method, with an increase in saturation (BS%) to 50% [13], the application rate would be 2.7 t ha$^{-1}$ for the 0–0.2 m layer and lime with ECCE of 77.6%. In this circumstance, soybean yields estimated by the sigmoid models used in the 2019/2020 and 2020/2021 crop years would result in soybean grain yields 46% and 45% lower than those obtained using a rate of 10 t ha$^{-1}$, respectively. Therefore, the average increase would be 32% (2019/2020) and 29% (2020/2021). It is worth noting that only from a rate of 10 t ha$^{-1}$ was it possible to achieve in the first crop year the base saturation recommended by the authors in the 0–0.2 m layer.

In a study evaluating lime and gypsum rates in an opening area in the Cerrado of Piauí state, Brazil [56], recommend rates of 8 to 10 t ha$^{-1}$ for soybean cultivation in the first year. However, the authors mention that satisfactory yields in the second year (2700 kg ha$^{-1}$) were obtained using lime rates greater than 6.3 t ha$^{-1}$ in a no-till system. The authors also mention that gypsum provided an increase in soybean yield, considering two harvests with limited amounts of rain (2015/2016) and irregular distribution of rain (2016/2017).

When collecting information on soil fertility management in western Bahia, consultants, and grain and fiber producers carry out soil analysis, but, in practice, they do not use such analyses to calculate the amount of lime, gypsum, and fertilizers to be applied. The region's soils have a predominantly sandy texture, in which rates of 3 and 1 t ha$^{-1}$ of lime and gypsum, respectively, would be the most recommended according to official criteria. In practice, however, farmers often use rates greater than 5–7 t ha$^{-1}$ of lime [57].

The use of high rates, therefore, guarantees an immediate correction of soil acidity, allowing soybean cultivation in the first year because a reaction of 20 to 40% of the lime applied at high rates is equivalent to the amount necessary to correct soil acidity in the order of magnitude recommended by official criteria and guarantees adequate soil fertility for soybean cultivation and greater yields in the first year [6].

## 5. Conclusions

Lime rates for high soybean grain yield in first-crop areas of the Matopiba region should not exceed 10 t ha$^{-1}$ or up to twice the value recommended by standard methods or official recommendation manuals. High lime rates (>10 t ha$^{-1}$) caused a decrease in the concentrations of P, K, and cationic micronutrients in soil, resulting in reductions in leaf contents of macro- and micronutrients in soybean plants. The combined use of gypsum and lime in areas recently converted from Cerrado to soybean cultivation provides a rapid chemical improvement in the soil profile, with a reduction in acidity components. The application of lime at the rate of 10 t ha$^{-1}$ is a feasible practice to improve the soil environment in the Matopiba region due to the short time available for chemical reactions to occur, allowing the cultivation of soybeans in Cerrado areas recently converted to agriculture.

**Supplementary Materials:** The following supporting information can be downloaded at: https: //www.mdpi.com/article/10.3390/agriculture14071034/s1, Figure S1: Experimental area at União Farm, Currais, Piauí, Brazil.; Figure S2: Data referring to Fe (a), Mn (b) and Zn (c) as a function of liming rates (e) and V (f) as a function of gypsum rates in the 2019/2020 cropping season, Fe (d) as a function of liming rates and Cu (g) as a function of gypsum rates in the 2020/2021 cropping season. Currais, Piauí, Brazil.; Figure S3: Leaf concentrations of N (a), P (b), K (c), Cu (d), Fe (e), Mn (f) and Zn (g) as a function of liming rates and of Zn (h) as a function of gypsum rates in soybean, in the 2019/2020 agricultural year. Currais, Piauí, Brazil.; Figure S4: Leaf concentrations of N (a), P (b), K (c), Mn (e), Fe (f) and Zn (g) as a function of liming rates and of K (g) and B (h) as a function of gypsum rates in soybean, in the 2020/2021 cropping season. Currais, Piauí, Brazil. Table S1: Average values, F test and coefficient of variation (CV) for soil chemical attributes (0.0–0.2 m) as a function of different liming and gypsum rates in soybean (2019/2020 cropping season) in a Cerrado area. Currais, Piauí, Brazil.; Table S2: Average values, F test and coefficient of variation (CV) for soil chemical attributes (0.2–0.4 m) as a function of different liming and gypsum rates in soybean (2019/2020 cropping season) in a Cerrado area. Currais, Piauí, Brazil.; Table S3: Average values, F test and coefficient of variation (CV) for soil chemical attributes (0.4–0.6 m) as a function of different liming and gypsum rates in soybean (2019/2020 cropping season) in a Cerrado area. Currais, Piauí, Brazil. Table S4: Average values, F test and coefficient of variation (CV) for soil chemical attributes (0.0–0.2 m) as a function of different liming and gypsum rates in soybean (2020/2021 cropping season) in a Cerrado area. Currais, Piauí, Brazil.; Table S5: Average values, F test and coefficient of variation (CV) for soil chemical attributes (0.2–0.4 m) as a function of different liming and gypsum rates in soybean (2020/2021 cropping season) in a Cerrado area. Currais, Piauí, Brazil.; Table S6: Average values, F test and coefficient of variation (CV) for soil chemical attributes (0.4–0.6 m) as a function of different liming and gypsum rates in soybean (2019/2020 cropping season) in a Cerrado area. Currais, Piauí, Brazil.; Table S7: Average values, F test and coefficient of variation (CV) for non-exchangeable soil Ca, Mg and Ca + Mg (0.0–0.2 m) as a function of different liming and gypsum rates in soybean (2019/2020 and 2020/2021 cropping seasons) in a Cerrado area. Currais, Piauí, Brazil.; Table S8: Average values, F test and coefficient of variation (CV) for the concentration of macro and micronutrients in the soybean diagnostic leaf as a function of liming and gypsum rates (2019/2020 and 2020/2021 cropping seasons) in a Cerrado area. Currais, Piauí, Brazil.; Table S9: Average values, F test and coefficient of variation (CV) for the soybean grain yield as a function of liming and gypsum rates (2019/2020 and 2020/2021 cropping seasons) in a Cerrado area. Currais, Piauí, Brazil.; Table S10: Average values (%) of residual limestone (non-exchangeable Ca and Mg) as a function of liming and gypsum rates in soybean at the 0.0–0.2 m soil layer in a Cerrado area. Currais, Piauí, Brazil.

**Author Contributions:** Conceptualization, J.J.d.J.L., T.P.R., R.M. and H.A.d.S.; data curation, D.B.d.O., J.J.d.J.L. and H.A.d.S.; formal analysis, D.B.d.O., A.P.C., K.G.B., A.C.G.M. and A.P.M.d.S.; funding acquisition, D.B.d.O., J.J.d.J.L., T.P.R., R.M. and H.A.d.S.; investigation, D.B.d.O., A.P.C., K.G.B., A.P.M.d.S., H.A.F.d.A. and P.M.C.; methodology, D.B.d.O., J.J.d.J.L., T.P.R. and H.A.d.S.; project administration, D.B.d.O. and J.J.d.J.L.; resources, D.B.d.O., J.J.d.J.L. and R.M.; software, D.B.d.O. and J.J.d.J.L.; supervision, D.B.d.O. and J.J.d.J.L.; validation, T.P.R. and R.M.; visualization, D.B.d.O. and E.S.; writing—original draft, D.B.d.O., J.J.d.J.L., C.A.F.d.S., J.O.L.d.O.J. and E.S.; writing—review and editing, D.B.d.O., J.J.d.J.L., E.S. and H.A.d.S. All authors have read and agreed to the published version of the manuscript.

**Funding:** This study was funded by Empresa Brasileira de Pesquisa Agropecuária—Embrapa Meio-Norte (Project: 30.20.90.038.00.00)/Aprosoja Piauí/Federal University of Piauí. The last author is supported by the fellowship CNPq-PQ (314920/2020-0).

**Data Availability Statement:** The raw data supporting the conclusions of this article will be made available by the authors upon request.

**Acknowledgments:** The authors would like to acknowledge Empresa Brasileira de Pesquisa Agropecuária—Embrapa Meio-Norte/Aprosoja Piauí/Federal University of Piauí—UFPI (30.20.90.038.00.00).

**Conflicts of Interest:** Carlos Antonio Ferreira de Sousa and Henrique Antunes de Souza were employed by Brazilian Agricultural Research Company. The remaining authors declare that the research was conducted in the absence of any commercial or financial relationships that could be construed as potential conflicts of interest.

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
