# Peer review of "Lime and Gypsum Rates Effects in New Soybean Areas in the Cerrado of Matopiba, Brazil"

_agriculture, doi:10.3390/agriculture14071034_

Round 1

Reviewer 1 Report

Comments and Suggestions for Authors

The manuscript evaluate the effects of high lime and gypsum rates in newly converted areas for soybeans production in the Cerrado, the work not novelty,but practical and meaningful.  I suggest the author should determine the soil buffering capacity and get the correlation with different paremeters.

Comments on the Quality of English Language

The quality of english language is good.

Reviewer 2 Report

Comments and Suggestions for Authors

Review

The authors in the manuscript titled "High lime and gypsum rates in new soybean areas in the Cerrado of Matopiba, Brazil" raises an important issue regarding mitigating the effects of soil acidification and high concentrations of toxic forms of aluminum through liming and the use of gypsum. However, after analyzing the text, I have a few comments and questions:

Ø  Please use the current literature on classification according to WRB-IUSS Working Group WRB. 2022. World Reference Base for Soil Resources. International soil classification system for naming soils and creating legends for soil maps. 4th edition. International Union of Soil Sciences (IUSS), Vienna, Austria

Ø  I do not understand why the authors provide the analysis results in units per volume (dm3), please provide the content, e.g. P and K in mg kg-1, sorption capacity in cmol kg-1, soil texture - fraction content in percent (%);

Ø  In my opinion, in order to fully illustrate the results, reaction results determined not only in the CaCl2 extract but also in the water extract are needed. This result will show us what the current amount of hydrogen is in the soil solution.

Ø  In line 172 it is written: "organic matter (OM) by wet digestion with potassium dichromate" I do not understand because with potassium dichromate the organic carbon content is determined, which can then be converted into OM content by multiplying by the factor 1.724. Please explain whether this is the organic carbon or organic matter content?

Ø  Moreover, in my opinion, for the general characteristics of soils, there is a lack of analyzes regarding the content of total nitrogen. Knowing the content of this component, we can calculate the carbon to nitrogen ratio, based on which we can draw conclusions about the quality of organic matter and biological activity of the soil,

In my opinion, the work requires corrections and additions.

Reviewer 3 Report

Comments and Suggestions for Authors

The author highlighted the role of high dozes of lime and gypsum in soybean. This study is interesting but have some major issues and need to be addressed before acceptance.

1. The title wording is not correct, why author use high lime and gypsum rates. Please correct it, why you use the word high? if you used high word it mean that you are testing high dose of fertilizer, and researchers are trying to find any other way to reduce the use of high doze. Please carefully check it.

2. Line 13 is wrong, (High dose of amendments?) which amendments?

3. please do not use the initial of the word captial except the start of sentence. LIne 13, Increasing?

4. Line 15 and 16, should be re-write as, different doses of lime and gypsum.

5. Line 29, The application of high lime rates (10 t ha-1 )? what do you think 10 is high?15 and 20 t ha-1 are not high? Please carefully read these mistakes and correct it.

6. Over all introduction is written well but not in proper sequecne also text lenght is more, Try to be specific and short.

7. Must give credit to all researchers methods (References), which you have included in the materials and methods section.

8. Line 211, Replace this (Soil fertility in the 2019/2020) crop years with (Soil fertility in the first growing season), and same for the second year.

9. In figure 2, why initial two levels of gypsum are missing in some charts?

10. Revise the scale for K content.

11. Please also consider comment number 9 for figure 3-6.

12. I must suggest to add correlation figures for all data, and thus reader will understand the trend of your data and work.

13. Discussion is well written however conclusion need to re-phrase and must be in one paragraphs.

Round 2

Reviewer 2 Report

Comments and Suggestions for Authors

Review 2

In my opinion, the authors have not sufficiently improved the submitted manuscript according to my suggestions. Below are my justifications:

1. The following argument is unconvincing to me "Regarding the units for chemical

analysis, the results were generated using the Brazilian standards for soil chemical analysis, as mentioned in the Material & Methods section - Reference 28 (Teixeira, P.C.; Donagemma, G.K.; Fontana, A.; Teixeira, W.G. Manual de Métodos de Análise de Solo; Embrapa: Brasília, 2017; ISBN 9788570357717.) and Reference 29 (Raij, B. van; De, A.J.C.; Cantarella, H.; Quaggio, J.A. Analálise Química Para Avaliação Da Fertilidade de Solos Tropicais; IAC.; Instituto Agronômico: Campinas, 2001; ISBN 85-85564-05-9.)” because the above methodology allows for the expression of content expressed in units in terms of volume, but such a solution is used only for soils (soil substrates) in horticultural crops, most often in greenhouses. We do not know the density of the analyzed soil, therefore the obtained results cannot be compared with the results obtained by other researchers. The presented arguments are insufficient for me. In my opinion, there is only such a possibility in the presented methodology, but it is not commonly used even in research conducted in Brazil.

2. In my opinion, it is necessary to provide pH in the water extract and CaCL2 extract. The arguments presented by the authors are unacceptable to me. Please provide additional results regarding this parameter. This approach will provide broader possibilities for interpreting the results.

3. If organic carbon was determined, please provide the results of this parameter, not the results from the conversion.

4. In my opinion, information on the content of total nitrogen is necessary. This parameter is one of the basic properties of soils. As I mentioned in the first review, it is possible to calculate the carbon to nitrogen ratio based on it and then draw a number of important conclusions on this basis. The authors' answers do not convince me. Please provide additional analyses regarding this parameter.

In my opinion, in its current form, the manuscript does not meet the requirements for publication. It requires corrections and additions

Reviewer 3 Report

Comments and Suggestions for Authors

The revised version has been significantly improved, therefore i suggest to accept at this stage

Author Response

We thank you for the valuable comments and suggestions, which contributed significantly to the improvement of the manuscript quality.